# Evaluation of the marginal and internal gaps in 3D-printed interim crowns using different finish line detection methods: An in vitro study

Nguyen Thi Thu Huong[1,2], Nguyen Phuong Thao[1], Nguyen Minh Duc[3,4‡],
Nguyen Viet Anh[5‡], Nguyen Thu Hang[1☯], Nguyen Thi Nhu Trang[1☯], Khieu Thanh Tung[6☯],
Nguyen Thu Tra[1]*

1 Department of Prosthodontics, School of Dentistry, Hanoi Medical University, Hanoi, Vietnam,
2 Department of Odonto-stomatology, Hanoi Medical University Hospital, Hanoi Medical University, Hanoi, Vietnam, 3 Department of Oral Surgery, School of Dentistry, Hanoi Medical University, Hanoi, Vietnam,
4 Division of Research and Treatment for Oral Maxillofacial Congenital Anomalies, Aichi Gakuin University, Nagoya, Japan, 5 Department of Prosthodontics, Faculty of Dentistry, Phenikaa University, Hanoi, Vietnam, 6 Department of Implantology, School of Dentistry, Hanoi Medical University, Hanoi, Vietnam

☯ These authors contributed equally to this work.
‡ These authors also contributed equally to this work
* nguyenthutra@hmu.edu.vn

## Abstract

### Objective

This study aimed to evaluate the accuracy of the Dentbird crown software in automatically detecting the finish line for interim crowns.

### Materials and methods

A mandibular first molar typodont model with a chamfer finish line was prepared and scanned ten times, resulting in ten STL files. The finish line for each file was detected using both automatic and semi-automatic methods in two software programs: the CEREC InLab system and Dentbird software. The internal and marginal gaps were measured at four locations: mesial, distal, buccal, and lingual- using the silicone replica technique. One-way analysis of variance (ANOVA) and post-hoc analyses ($\alpha = 0.05$) were performed to detect statistical differences in the marginal and internal gaps among the groups.

### Results

The results revealed significant differences in internal and marginal gaps between the automatic methods of the Dentbird software and the CEREC system ($p < 0.05$). However, no significant differences were found in the semi-automatic methods between the two systems ($p > 0.05$). Although the fits of crowns automatically designed by Dentbird software were inferior to those of the semi-automatic method by Dentbird

**Data availability statement:** All relevant data are within the manuscript and its Supporting Information files.

**Funding:** The author(s) received no specific funding for this work.

**Competing interests:** NO authors have competing interests.

software and the CEREC In Lab system, the values of all four groups were within the clinically acceptable range (<120 μm).

## Conclusion

The internal and marginal fit of crowns designed using the automatic and semi-automatic modes in Dentbird, a freely available CAD platform, fell within the range of traditional clinical acceptability. Hence, automatically generated crowns may be considered appropriate for immediate provisional applications, the semi-autonomic finishing line detection can be used for long term crowns in clinical practice.

---

## Introduction

Marginal adaptation is a key factor in the success of dental restorations, as it influences the long-term prognosis of dental crowns [1–5]. A marginal gap is defined as the vertical distance between the finish line of the preparation and the cervical margin of the restoration [6,7]. Poor marginal adaptation of crown can cause hypersensitivity, secondary caries, gingivitis, and periodontal problems [4,5,8,9]. Large marginal discrepancies also result in a thick cement film, exposing the luting material to the oral environment and decrease of the longevity of the prosthetics restorations. Moreover, the internal fit is also an important factor for the success of dental crowns. The term internal gap corresponds to the vertical distance measured between the axial wall of the preparation and the surface of a crown [6,7]. Internal fit is directly associated with crowns retention and resistance properties [8]. Therefore, the study of factors that affect marginal and internal gaps is critically important in restorative dentistry.

During the last two decades, production stages are increasingly becoming automated in dentistry. Producing dental restorations with CAD/CAM technology s become more common in recent years [10,11] and provided dentists and dental technicians many benefits such as saving time, increased quality, and improved accuracy of restorations [12].

In term of marginal adaptation, there are three common methods that are used in detecting the finish line. In the manual method, operators detect the finish line by drawing a continuous line on the tooth preparation model. In the automatic method, the finish line is automatically detected by dental CAD software programs without human guidance. In the semi-automatic method, after automatic detection, the operator carefully checked and adjusted the finish line. Although the automatic methods can reduce manual operations, the accuracy of finish line detection, since now, is mostly limitedly evaluated in charged software rather than free platforms.

There are previous studies that evaluated the quality of restorations designed by digital software [13–18]. The results of these studies show that the restorations designed by CAD/CAM software have acceptable marginal adaptability [12,19–21]. This is a limitation because these are all paid software and need to be installed on the computer. Free-charged software, like Dentbird crown software (Imagoworks, Seoul, South Korea)a in which users can access it online without installing the

program, provided a revolution in simplifying the process in designating single unit restorations. Howeverr, limited evidence is available on the marginal gap and the internal fit of the crown designed by these platforms as they are still fairlynew. Therefore, this in-vitro study aims to evaluate the marginal gap and internal gap of the crown automatically and semi-automatically designed by Dentbird crown software. The null hypothesis was that no difference would be found in the marginal and internal gap of the crowns designed by Dentbird with a well-known charged CAD software.

## Materials and methods

### 2.1. Fabrication of the master models

In this study, a typodont model of the mandibular first molar was designed with a height of 5.0 mm, mesiodistal width of 11.0 mm, buccolingual width of 10.0 mm, shoulder margin width of 1.2 mm, and axial wall taper of 6° (Fig 1). To completely avoid light reflectance during scanning, we did not polish the typodont to a glossy level, smooth but not shiny.

A power analysis was designed to obtain sufficient power for a statistical test of the null hypothesis that there is no difference between the tested groups. Assuming an alpha (α) level of 0.05 (5%), a beta (β) level of 0.2 (power = 80%), and an effect size of 0.8 calculated based on the results of a previous study [22], the predicted sample size (n) was found to be 16 samples, 8 samples per group. We decided to collect 10 samples per group to prevent experimental variability. Sample size calculation was performed using G*Power version 3.1.9.4 (University of Düsseldorf, Germany) [23].

### 2.2. Detection of the finish line of tooth preparation in computer software

The tooth model was scanned ten times using the CEREC Primescan scanner (Sirona, Bensheim, Germany), producing ten stereolithography (STL) files. These files were imported into two dental CAD software programs to detect the finish line. In the CEREC InLab system (Sirona, Bensheim, Germany), Groups A and B were created, with Group A using the semi-automatic method and Group B the automatic method. In Dentbird software (Imagoworks, Seoul, South Korea), Groups C and D were created, with Group C using the semi-automatic method and Group D the automatic method (Fig 2).

In the automatic method, the finish line was detected based on one or two guidance points. Firstly, a point is selected from the scanned image by the operator. Calculate the curvature of the area around the point and adjust the starting point

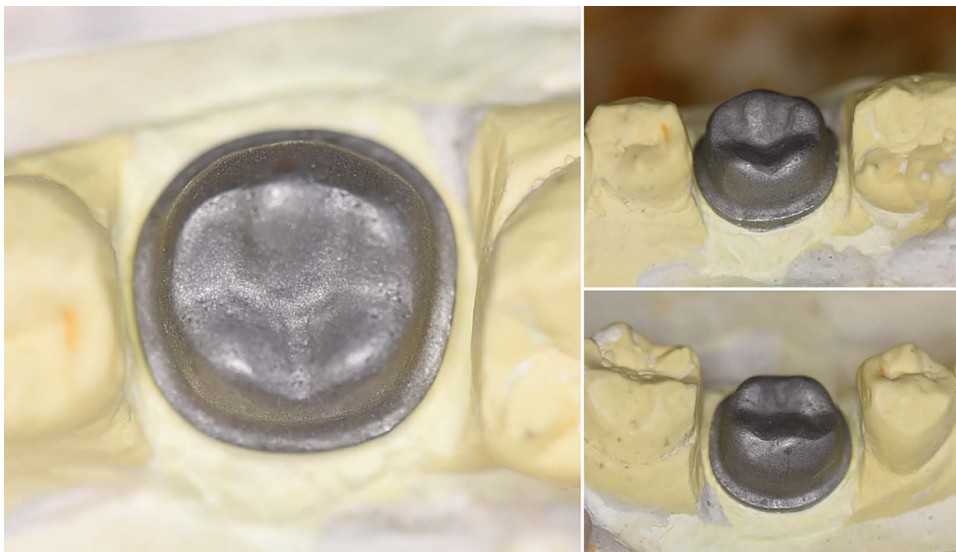

**Fig 1. Master die.**

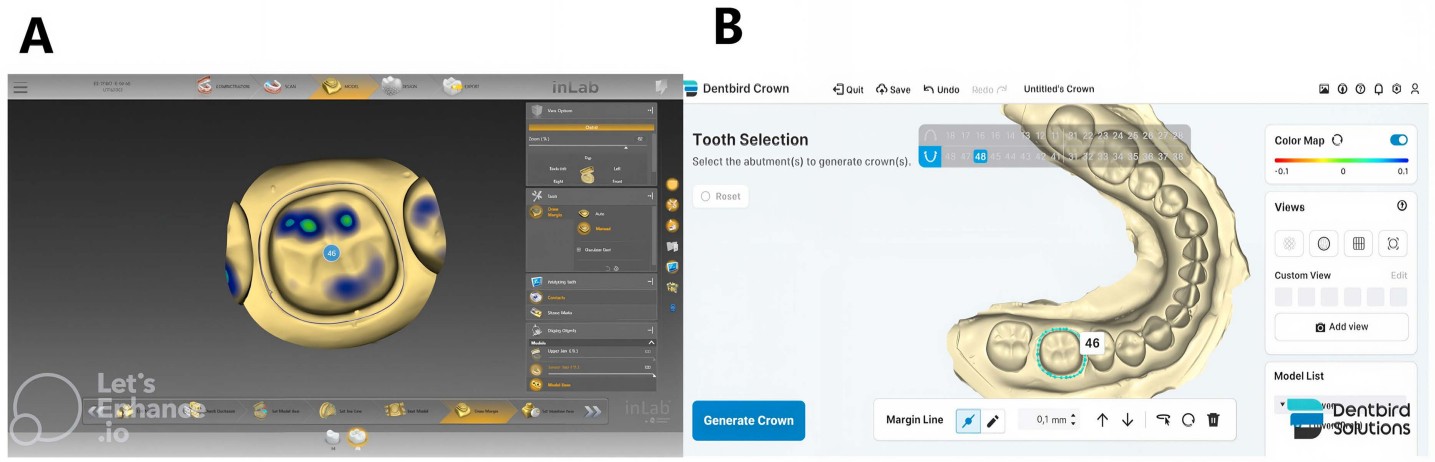

**Fig 2. Finish line registration of a tooth preparation scan model. (A)** CEREC InLab System, **(B)** Dentbird software.

to where the curvature changes the most. Margin is then found by computing the curvature of adjacent meshes and tracing the edge with the greatest change in curvature. This process is ended by returning to the initial set point. Conversely, in the semi-automatic method, after automatic detection, the operator reviewed and manually fine-tuned the finish line based on their observations of its path.

Based on the registered finish line, 40 virtual copings (n = 10 per group) were designed and saved in stereolithography (STL) file format. All finish line registrations and coping design procedures were carried out by a skilled operator who was blinded to the study's purpose. Subsequently, all resin crowns were printed using Dio Probo Z 3D printer (Dio Co., Busan, Korea) and DIOnavi-C&B ink for temporary restorations (Dio Co., Busan, Korea) following the crown printing option available in the 3D printer, with a 30 μm cement gap starting 1 mm from the finish line margin [16,24]. The build orientation was set at 15 degrees relative to the build plate, with the crown's occlusal surface facing upward. Each printing layer had a thickness of 50 μm, and one crown was printed in approximately 15 minutes. The printed crown was then washed in isopropyl alcohol (IPA) and post-cured for 60 minutes, following the manufacturer's recommendations and the protocol described by Dohyun Kim [25].

### 2.3. Internal and marginal gap evaluation (Silicone replica technique SRT)

The silicone replica technique was used to measure both the internal and marginal gaps of all crowns. Light-body silicone impression material (Elite HD +; Zhermack, Rovigo, Italy) was mixed and applied to the crown. The crown was then placed on the prepared abutments and subjected to a 5 kg load applied using a calibrated weight to ensure consistent pressure for 3 minutes until the silicone material set. This standardized loading procedure was performed to ensure uniform application of pressure across all samples. After polymerization, the crown was carefully removed, leaving a silicone impression on the abutment that accurately represents the gap between the crown and the prepared tooth. The heavy-body silicone (Elite HD +; Zhermack, Rovigo, Italy) was then applied over the remaining light impression on the abutment to provide support to the light body silicone for another three minutes.

After setting, the silicone replica was removed and sectioned at the midline in bucco-lingual and mesial-distal directions using surgical blade no.15. To standardize the four measurement points on the replica silicon (mesial, distal, buccal, lingual), we first draw a central fit line through the central fit, intersecting the mesial and distal walls at two crossing points. Next, we draw a line that is perpendicular to the central fit line, through the lowest point of central fossa and meet the lingual and buccal walls at two other crossing points. Then, draw four lines from above crossing points along the typodont

axis, meet the finishing line at the mesial, distal, lingual and buccal points, respectively. All the silicon replicas were sectioned by two lines connecting the mesial–distal and buccal–lingual points to obtain standardized cross-sectional planes for analysis (Fig 3). The thickness of the light body at four points was observed using a Stereo microscope (TERINO 1200X-HD) at a magnification of 100x with a digital camera. The marginal and internal gaps were measured at 4 points using image analysis software Image J (version 1.46, Java) (Fig 4). All measurements were done by a technician who is master in using microscope and Image J, also blind from the study design and objectives.

## 2.4. Statistical analysis

The mean and standard deviation of the internal and marginal gap measurements were calculated. The data were analyzed with a statistical software program (IBM SPSS Statistics, v25.0; IBM Corp, Chicago, IL, USA). Results were compared using a one-way analysis of variance (ANOVA) and the post hoc Tukey's test at a significance level of 0.05.

## 3. Results

The test pf homogeneity of variances confirmed that measured values at marginal and internal gaps at buccal, lingual, mesial, distal points, RMS internal and margianl gaps are homogennous. The mean and standard deviation values with 95% confident interval of the internal and marginal areas are listed in Table 1.

The descriptive statistics showed that in general, Group A recorded the least marginal and internal gaps while Group D had the highest marginal and internal gaps as compared with the other groups (Table 1).

Comparison of the overall marginal and internal gaps among the different groups using one-way ANOVA test revealed a statistically significant difference among groups ($p < 0.05$). The post-hoc analysis revealed that the values of group D were significantly higher than those of the other three groups ($p < 0.05$). However, no significant differences were found between group C, A and B ($p > 0.05$) (Table 2).

Consistent with the overall internal and marginal gap findings, subgroup analyses at the four measured points showed that group A again exhibited the lowest values, whereas group D had the highest (Table 3). However, a different trend was observed for the marginal gap data. Although groups A, B, and C demonstrated comparable internal gaps across all measured points, the marginal gaps in group A were significantly smaller than those in groups B and C on the buccal side, and smaller than those in group C on the mesial and distal sides (Table 4).

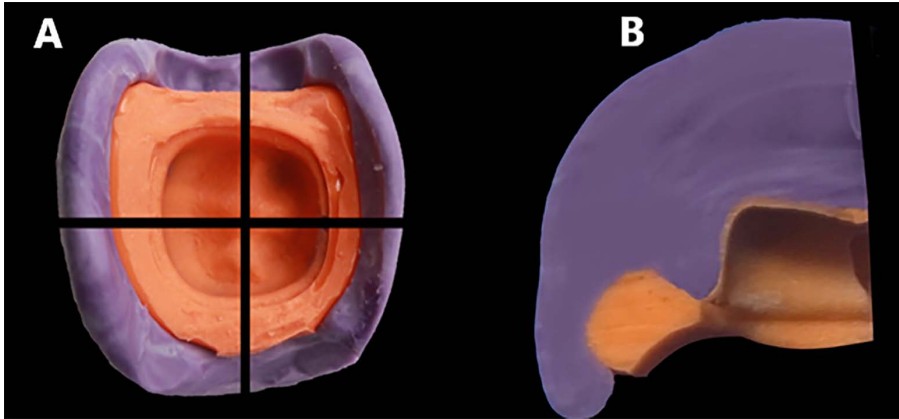

**Fig 3. (A) Four sections of silicone replica, (B) Replica after segmentation.**

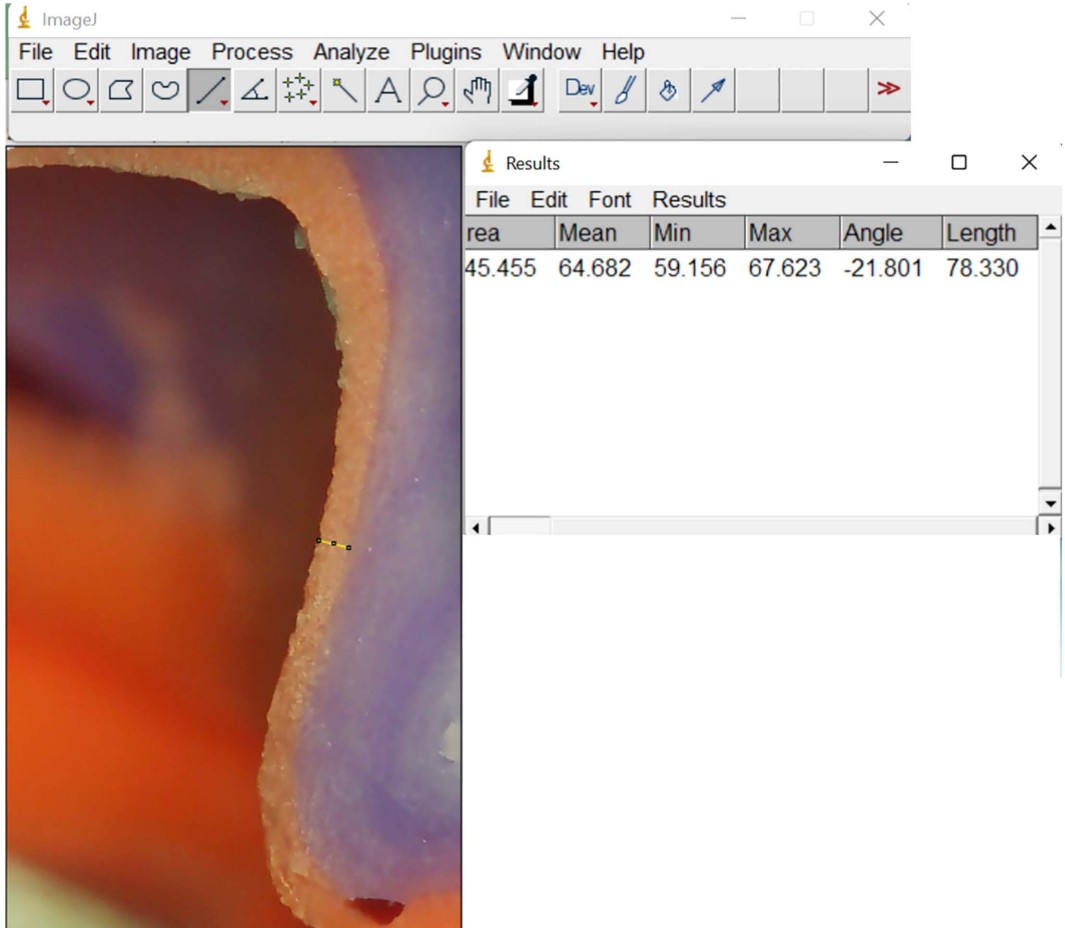

**Fig 4. Replica technique with silicon in ImageJ.**

**Table 1. Mean and standard deviation values (95% confident interval) of the internal and marginal gaps among the different groups (unit: μm).**

| Area | CEREC InLab system | | Dentbird software | |
| --- | --- | --- | --- | --- |
| | **Group A Semi-automatic method** | **Group B Automatic method** | **Group C Semi-automatic method** | **Group D Automatic method** |
| Internal gap | 58.73±5.11[a] (54.80 - 62.65) | 63.99±6.87[a] (56.06 - 71.91) | 66.05±6.73[a] (60.88–71.22) | 78.10±9.44[b] (70.84–85.36) |
| Marginal gap | 74.52±6.62[a] (69.43–79.61) | 78.91±8.95[a] (72.03–85.79) | 81.97±5,9[a] (77.43–86.50) | 116.64±9,61[b] (109.25–124.02) |

*According to the One-way ANOVA test, $P < 0.05$

In each column, values followed by same letter are statistically similar ($p > 0.05$)

**Table 2. Post-hoc analysis of overall internal and marginal gaps among the four groups.**

| | Internal gap | | | Marginal gap | | |
| --- | --- | --- | --- | --- | --- | --- |
| | **Group A** | **Group B** | **Group C** | **Group A** | **Group B** | **Group C** |
| Group B | 0.181 | – | – | 0.249 | – | – |
| Group C | 0.066 | 0.596 | – | 0.055 | 0.419 | – |
| Group D | 0.000* | 0.001* | 0.004* | 0.000* | 0.000* | 0.000* |

**Table 3. Internal and marginal gaps in buccal, lingual, mesial, distal points among the four groups.**

| | | Group A | Group B | Group C | Group D |
|---|---|---|---|---|---|
| Internal gap | Buccal | 57.34±2.43 | 63.90±3.44 | 63.86±2.32 | 75.43±3.05 |
| | Lingual | 58.36±1.16 | 63.99±3.44 | 65.43±2.29 | 76.92±3.06 |
| | Mesial | 58.07±1.78 | 63.99±3.44 | 66.66±2.25 | 78.77±3.15 |
| | Distal | 60.94±1.79 | 63.99±3.44 | 68.06±2.09 | 81.15±3.33 |
| Marginal gap | Buccal | 58.72±1.70 | 75.60±3.02 | 79.41±1.96 | 114.76±3.49 |
| | Lingual | 74.52±2.21 | 77.54±3.02 | 80.73±1.96 | 115.91±3.71 |
| | Mesial | 74.52±2.21 | 79.80±2.97 | 82.72±1.94 | 116.29±2.81 |
| | Distal | 74.52±2.21 | 82.33±2.95 | 84.90±2.01 | 119.31±3.69 |

**Table 4. Post-hoc analysis of internal and marginal gaps in buccal, lingual, mesial, distal points among the four groups.**

| | | Internal gap | | | Marginal gap | | |
|---|---|---|---|---|---|---|---|
| | | Group A | Group B | Group C | Group A | Group B | Group C |
| Buccal | Group B | 0.108 | – | – | 0.000* | – | – |
| | Group C | 0.115 | 0.974 | – | 0.000* | 0.316 | – |
| | Group D | 0.000* | 0.008* | 0.007* | 0.000* | 0.000* | 0.000* |
| Lingual | Group B | 0.14 | – | – | 0.452 | – | – |
| | Group C | 0.067 | 0.702 | – | 0.128 | 0.429 | – |
| | Group D | 0.000* | 0.002* | 0.004* | 0.000* | 0.000* | 0.000* |
| Mesial | Group B | 0.136 | – | – | 0.134 | – | – |
| | Group C | 0.034 | 0.496 | – | 0.028 | 0.447 | – |
| | Group D | 0.000* | 0.001* | 0.004* | 0.000* | 0.000* | 0.000* |
| Distal | Group B | 0.441 | – | – | 0.057 | – | – |
| | Group C | 0.078 | 0.305 | – | 0.013 | 0.521 | – |
| | Group D | 0.000* | 0.000* | 0.002* | 0.000* | 0.000* | 0.000* |

## 4. Discussion

The purpose of this study was to evaluate the marginal and internal gap of the crown with the finish line automatically designed by Dentbird crown software. The current study used the 2D silicon replica technique to evaluate the marginal and internal fit of four groups. Previous studies utilized silicone replica techniques because it is a non-destructive and high-reliability method [26–29].

According to our current findings, there were statistically significant differences between the four groups in overall marginal and internal s. Thus, the initial null hypothesis was rejected.

Significant differences were found between group D and the other three groups (C, A, B), the fit of crowns with finish line automatically designed by Dentbird software were inferior to those of the semi-automatic method by Dentbird software and the CEREC system in both automatic and semi-automatic methods. According to opinions supposed in 1970s regarding the clinically permissible range of marginal and internal gaps, < 120 μm is considered the clinically acceptable range suggested by most researchers [29–32]. However, the clinically acceptable range of the gaps remains inconsistent. For both cast and cemented restorations and CAD/CAM crowns, reported mean marginal and internal gaps range from 50 to 120 μm and 100–160 μm, respectively [33–36]. Particularly, an approximately 80 μm marginal gap is fracture resistant and unable to be detected visually or tactilely across all types of restorations [32,37]. In the present study, the overall internal gap values of all crowns were below 80 μm, which is well within the clinically acceptable range. Although the overall marginal gap of the Dentbird autonomic finishing-line detection group remained within the traditional acceptable

threshold, it was significantly higher than those observed in other groups and also higher than the level of 80 μm. These findings indicate that the Dentbird software with automatic finish line detection offers a time-efficient approach to designing temporary restorations with clinically acceptable fit. However, for long-term restorations, the semi-automatic finishing-line detection method should be preferred to ensure superior marginal adaptation and long-term clinical performance. Indeed, no significant differences between groups C, A, and B in both internal and marginal gaps show the reliability of detecting the finish line by semi-automatic method in Dentbird software compared to a well known Computer Aided Design software as Cerec Inlab system.

In terms of the gaps in each measured points, the marginal gaps in group A were significantly smaller than those in groups B and C on the buccal side, and smaller than those in group C on the mesial and distal sides, suggesting the advantages of semi-autonomic detection methods even in charged software [24,38]. In the present study, when automatic finish line detection was used, significant differences were found in the registration accuracy between the two software programs. This is because the automatic method was carried out based on the arithmetic interpretation of the software algorithm, which largely depends on the geometric differences of the model surfaces [18]. Thus, an error may occur when a geometrically indistinct area is reached during edge detection. Based on the working principle, it is very important to create a difference between prepared and unprepared tooth surfaces. The automatic method with the advantage of saving time can be used in cases where less accuracy is required, such as making temporary restoration for patients while waiting for the official restoration.

The semi-automatic method was additionally supported by the perception of the overall situation and prior clinical knowledge of the operator, which allows manual correction of registration errors in unfavorable or indistinct tooth preparation finish line conditions.[12] Therefore, in cases where high accuracy is required, it is recommended to use the semi-automatic method instead of the automatic method in CAD software.

To control the variables in the experiment, this study used a stainless-steel model to decrease the surface chipping. The silicon was always mixed in the ratio, ensuring the correct mixing time and the manufacturer's setting time. The pressure on the crowns was controlled to maintain an equal force between the groups. The silicone replica was carefully sectioned using surgical blade no. 15 to reduce the possibility of deformation and tearing of the impression material. All processes were carried out under the same conditions at a room temperature of 20°-22°C by. To control the bias, our study hired an operator who is expert in replica silicon, and a technician who is expert in using microscope and Image J software. Both were blind from all design and purpose of the study.

The main limitations of this study stem from its in vitro design, which cannot fully replicate clinical conditions. The crown fit was assessed only in the vertical plane on a single tooth—the mandibular first molar—with a shoulder, supragingival margin. Evaluating the fit in both horizontal and vertical dimensions, as well as on different tooth types and finish line configurations, would provide a more comprehensive understanding. Future research should therefore explore more complex clinical situations to enhance the applicability of these findings.

## 5. Conclusion

Within the limitations of this in vitro study, it was concluded that the mean internal and marginal gap values of crowns designed using Dentbird software were within the traditional clinically acceptable range (<120 μm) for both automatic and semi-automatic methods. However, crowns designed automatically by Dentbird software exhibited inferior fit compared to those designed using the semi-automatic method in Dentbird software and the CEREC InLab System. Notably, the fit of crowns designed with the semi-automatic method in Dentbird software was comparable to that of the CEREC InLab System. This suggests that Dentbird software with automatic finishing line detection can save time and reduce costs in designing temporary restorations while maintaining an acceptable fit.. For long-term crowns, using the software with semi-automatic finishing line detection provides an equivalent vertical fit compared to charged software.

## Supporting information

**S1 File. Raw data.**
(XLSX)

## Author contributions

**Conceptualization:** Thu Tra Nguyen.

**Data curation:** Nguyen Thi Thu Huong, Nguyen Phuong Thao.

**Funding acquisition:** Nguyen Minh Duc, Nguyen Thu Hang, Nguyen Thi Nhu Trang, Khieu Thanh Tung.

**Investigation:** Nguyen Thi Thu Huong, Nguyen Thu Hang.

**Methodology:** Nguyen Viet Anh.

**Resources:** Nguyen Thi Nhu Trang, Khieu Thanh Tung.

**Supervision:** Thu Tra Nguyen.

**Writing – original draft:** Nguyen Thi Thu Huong, Nguyen Phuong Thao.

**Writing – review & editing:** Thu Tra Nguyen, Nguyen Minh Duc, Nguyen Viet Anh.

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
