## [Decision Letter · Decision Letter 0]

14 Aug 2025

Dear Dr. Nguyen,

We look forward to receiving your revised manuscript.

Kind regards,

Nour Ammar

Academic Editor

PLOS ONE

Journal Requirements:

https://www.jstage.jst.go.jp/article/jpr/67/1/67_JPR_D_21_00344/_pdf/-char/en

In your revision ensure you cite all your sources (including your own works), and quote or rephrase any duplicated text outside the methods section. Further consideration is dependent on these concerns being addressed.

3. Please upload a copy of Figure 3 and Figure 4, to which you refer in your text on page 6. If the figure is no longer to be included as part of the submission please remove all reference to it within the text.

Additional Editor Comments :

Please carefully consider the reviewers' comments, particularly those relating to the clinical generalizability and limitations of the presented results.

Reviewer's Responses to Questions

**Comments to the Author**

1. Is the manuscript technically sound, and do the data support the conclusions?

Reviewer #1: Yes

Reviewer #2: Yes

Reviewer #3: Partly

2. Has the statistical analysis been performed appropriately and rigorously?

Reviewer #1: Yes

Reviewer #2: Yes

Reviewer #3: No

3. Have the authors made all data underlying the findings in their manuscript fully available?

Reviewer #1: Yes

Reviewer #2: Yes

Reviewer #3: Yes

4. Is the manuscript presented in an intelligible fashion and written in standard English?

Reviewer #1: Yes

Reviewer #2: Yes

Reviewer #3: Yes

Reviewer #1: I would like to thank you for your submission entitled "Evaluation of the Marginal and Internal Gaps in 3D-Printed Interim Crowns Using Different Finish Line Detection Methods: An In Vitro Study." The topic is clinically relevant and timely, especially considering the increasing interest in digital dentistry and the growing accessibility of free CAD software solutions like Dentbird.

I would like to respectfully recommend minor revisions before publication to enhance clarity and scientific depth:

Technical Clarification on Dentbird's Algorithm: Since your results point to the suboptimal performance of Dentbird's automatic finish line detection, it would strengthen your discussion to elaborate—if possible—on the underlying technical limitations of this algorithm. A clearer understanding of why this method underperforms compared to others will be beneficial to the reader.

Generalizability of Findings: As only a single tooth type (mandibular first molar) and one type of finish line (chamfer) were tested, the applicability of results to more complex clinical cases remains uncertain. It would be helpful to explicitly acknowledge this in your discussion.

Horizontal Fit Considerations: You have assessed vertical internal and marginal gaps, but did not evaluate the horizontal aspect of fit, which also plays an important role in clinical longevity. A short mention of this limitation and possible future directions would be appreciated.

Clinical Relevance: While your conclusions rightfully state that all gaps are within acceptable clinical limits, further reflection on clinical use cases—e.g., suitability of Dentbird's automatic method for short-term temporary crowns versus long-term restorations—could improve the practical value of your findings.

Reviewer #2: Title: Title is appropriate.

Keywords: Appropriate

Abstract: Details the main purpose and is concise.

Introduction: There are previous studies (ADD MORE RECENT REFERENCES) that evaluated the quality of restorations designed by digital software such as CEREC (13-15), Lava (13, 16), EXOCAD (12). The results of these studies show that the restorations designed by CAD/CAM software have acceptable marginal adaptability.

Methodology: Concise and detailed

Results: Well explained and easy to understand

Discussion: Concise

Conclusion: Meets the objective.

References: Incorporate 2 or more recent references as mentioned above.

Reviewer #3: Some questions and comments about your manuscript:

Abstract

The abstract is concise but could better emphasize the study's implications for clinical practice.

Introduction

This section effectively sets the context for marginal and internal gaps' importance but could integrate more recent literature on CAD/CAM automation.

What specific algorithms or geometric analyses differentiate automatic from semi-automatic detection, and why might Dentbird's approach lead to inconsistencies?

The null hypothesis is stated, but how does it align with prior evidence suggesting software-dependent variability in crown fits?

Beyond marginal adaptation, how do internal gaps specifically affect retention, and what evidence supports the <120 µm clinical threshold cited later? Is this level of adaptation still really acceptable today?

Materials and Methods

The methods are replicable but lack precision in some areas, such as software versions or exact measurement protocols.

How was the typodont model's dimensions (e.g., 6° taper) validated to mimic clinical preparations, and were any pilot tests conducted to ensure scanner accuracy?

In the silicone replica technique, what criteria were used to select the four measurement points (mesial, distal, buccal, lingual), and how was inter-observer reliability assessed if measurements were done by one person?

The power analysis references a prior study (ref. 17), but what effect size was assumed, and why was the sample size adjusted to 10 per group from the calculated 8?

For 3D printing, what resin material was used, what were the critical printing parameters (angle, layer, speed, etc.) and how were post-processing steps (e.g., curing time) standardized to minimize variability?

Results

Why are only means and standard deviations reported without confidence intervals or effect sizes, which could better illustrate the magnitude of differences?

The post-hoc analysis shows no differences between groups A, B, and C, but how do these compare regionally (e.g., buccal vs. lingual gaps) – were subgroup analyses performed?

Table 1 uses superscript letters for similarity, but what p-values correspond to these comparisons, and was homogeneity of variance confirmed for ANOVA?

Discussion

Here I suggest that you more critically approach limitations and alternative explanations.

Given the inferior fit in Dentbird's automatic mode, what software-specific factors (e.g., algorithm sensitivity to curvature) might explain this, and how could future updates address it?

The study cites clinical acceptability (<120 µm), but how do these gaps compare to in vivo studies where oral fluids or cementation might alter fits?

Again, is this level of adaptation really acceptable today?

Limitations mention in vitro constraints, but why not discuss potential biases from using a metal typodont (e.g., vs. natural tooth reflectance in scanning)?

Your study is confined to a single tooth type (mandibular first molar) with a chamfer finish line, limiting generalizability. No evaluation of horizontal gaps or more complex clinical scenarios. Please comment.

Conclusion

How might these results influence the choice between free and paid software in educational or low-resource settings?

The emphasis on semi-automatic equivalence between software is strong, but what training implications does this have for operators relying on automation?

Figures

Fig. 1 could show different views of the prepared tooth.

Fig. 2, as presented, lacks the resolution that allows for better viewing of the software screens.

References

All references must adhere to PLOS ONE standards. Please check carefully.

Some references may be incomplete.

Language

There are some typos and inconsistencies through the text, which deserve careful review.

**Do you want your identity to be public for this peer review?** For information about this choice, including consent withdrawal, please see our Privacy Policy

Reviewer #1: No

Reviewer #2: No

Reviewer #3: No

---

## [Author Response · Author response to Decision Letter 1]

26 Nov 2025

Responses to reviewers

We sincerely thank you for your thorough review and insightful comments. We have carefully addressed each comment below and revised the manuscript accordingly. Changes made in the manuscript are highlighted using Track Changes. The line numbers and page numbers mentioned refer to the revised version of the manuscript.

1. Please ensure that your manuscript meets PLOS ONE's style requirements, including those for file naming. The PLOS ONE style templates can be found.

Response: Thank you. We have checked and carefully corrected all the headings, figure and table style names.

2. We noticed you have some minor occurrence of overlapping text with the following previous publication(s), which needs to be addressed: In your revision ensure you cite all your sources (including your own works), and quote or rephrase any duplicated text outside the methods section. Further consideration is dependent on these concerns being addressed.

Response: Thank you for your warning. We did change the introduction part and removed the overlapped sentences. Actually, we did not focus on the software algorithms. We would like to just test some fit values of the crowns that were designed by a free software.

3. Please upload a copy of Figure 3 and Figure 4, to which you refer in your text on page 6. If the figure is no longer to be included as part of the submission please remove all reference to it within the text.

Response: Thank you. We added the cite places of Figure 3 and 4 into the main text (Lines 144 and 147 page 7).

4. Please include captions for your Supporting Information files at the end of your manuscript, and update any in-text citations to match accordingly. Please see our Supporting Information guidelines for more information:

Response: Thank you. We corrected the name of supporting file to S1 Dataset

Reviewer's Comments to the Author

Reviewer #1:

1. Technical Clarification on Dentbird's Algorithm: Since your results point to the suboptimal performance of Dentbird's automatic finish line detection, it would strengthen your discussion to elaborate—if possible—on the underlying technical limitations of this algorithm. A clearer understanding of why this method underperforms compared to others will be beneficial to the reader.

Response: Thank you. We have adjusted the parts mentioning the software algorithms since this study does not focus on evaluating the software algorithms. Also, the algorithms of most CAD software are not published yet. Our study suggested the suboptimal performance of Dentbird in both autonomic and semi-autonomic finishing line detection. It might be because of the greater and more accumulated learning data throughout the R&D process of Cerec Inlab rather than the software algorithms.

2. Generalizability of Findings: As only a single tooth type (mandibular first molar) and one type of finish line (chamfer) were tested, the applicability of results to more complex clinical cases remains uncertain. It would be helpful to explicitly acknowledge this in your discussion.

Horizontal Fit Considerations: You have assessed vertical internal and marginal gaps, but did not evaluate the horizontal aspect of fit, which also plays an important role in clinical longevity. A short mention of this limitation and possible future directions would be appreciated.

Response: Thank you. We have added these limitations into the discussion part: Line 234 to 239 (page 13).

3. Clinical Relevance: While your conclusions rightfully state that all gaps are within acceptable clinical limits, further reflection on clinical use cases—e.g., suitability of Dentbird's automatic method for short-term temporary crowns versus long-term restorations—could improve the practical value of your findings.

Response: Thank you. We have added a discussion about a suitable application of Dentbird’s automatic methods into line 201 to 205 (page 11).

Reviewer #2:

1. Introduction: There are previous studies (ADD MORE RECENT REFERENCES) that evaluated the quality of restorations designed by digital software such as CEREC (13-15), Lava (13, 16), EXOCAD (12). The results of these studies show that the restorations designed by CAD/CAM software have acceptable marginal adaptability.

2. References: Incorporate 2 or more recent references as mentioned above.

Response: Thank you. We have added more recent papers evaluating the quality of crowns designed by charged CAD software (Line 66, page 11).

Reviewer #3:

1. Abstract: The abstract is concise but could better emphasize the study's implications for clinical practice.

Response: Thank you. We have added the detailed implications into conclusion of Abstract (Line 34 – 38, page 2).

2. Introduction

This section effectively sets the context for marginal and internal gaps' importance but could integrate more recent literature on CAD/CAM automation.

What specific algorithms or geometric analyses differentiate automatic from semi-automatic detection, and why might Dentbird's approach lead to inconsistencies?

Response: Thank you. As mentioned above, this study does not focus on evaluating the software algorithms or geometric analysis. Also, the algorithms of Dentbird and Cerec Inlab have not been published. We have deleted the sentences about software algorithms. The inferiorIt might be because of the greater and more accumulated learning data throughout the R&D process of Cerec Inlab rather than the software algorithms.

3. The null hypothesis is stated, but how does it align with prior evidence suggesting software-dependent variability in crown fits? Beyond marginal adaptation, how do internal gaps specifically affect retention, and what evidence supports the <120 µm clinical threshold cited later? Is this level of adaptation still really acceptable today?

Response: In this study, the null hypothesis is the fit of automatic and semi-automatic finishing line detection of Dentbird is not different with Cerec Inlab, a well-known CAD software. Even when previous studies reported the software-dependent variability in crown fits, if the variability is not significantly different in either semi-automatic or automatic finishing lien detection way, it would be good to report a cheaper and faster approach in dental crown fabrication.

Previous studies have reported that a more narrow internal gap will provide a higher bond strength between crown and abutment.

Regarding the level of 120 microns for clinical acceptance, we have discussed more in more detail in Line 189-209, page 11, discussion part. We did remove the level of marginal and internal gaps in introduction to avoid replication. Thank you very much for your deep comments.

4. Materials and Methods: The methods are replicable but lack precision in some areas, such as software versions or exact measurement protocols.

How was the typodont model's dimensions (e.g., 6° taper) validated to mimic clinical preparations, and were any pilot tests conducted to ensure scanner accuracy?

Response: Thank you for your comments. In clinical preparations, the convergence of the crown ranges from 6 to 12 degrees and no undercut in the prepared tooth. The typodont have been scanned and check for the convergence, undercut and smoothness in the CAD software before being used. The software, ImageJ, is basic and popular in laboratory research. The measuring function of ImageJ is the same among version since it is a very basic function. Pilot study evaluating the scanner accuracy has not been done. We just calibrate following the instructions from the company to ensure the accuracy of the scanned file as parameters announced by the manufacturer.

5. In the silicone replica technique, what criteria were used to select the four measurement points (mesial, distal, buccal, lingual), and how was inter-observer reliability assessed if measurements were done by one person?

Response: Thank you for pointing out a very important point. We have added the protocol how to detect four measured points into Line 131-139, page 7.

6. The power analysis references a prior study (ref. 17), but what effect size was assumed, and why was the sample size adjusted to 10 per group from the calculated 8?

Response: Thank you. We have added the size effect into Line 85, page 5. We adjusted the sample size from 8 to 10 to prevent the experimental variability and errors.

7. For 3D printing, what resin material was used, what were the critical printing parameters (angle, layer, speed, etc.) and how were post-processing steps (e.g., curing time) standardized to minimize variability?

Response: Thank you. We have added the material, printing parameters, and post-processing steps into the Lines 108-115, page 6.

8. Results

Why are only means and standard deviations reported without confidence intervals or effect sizes, which could better illustrate the magnitude of differences?

The post-hoc analysis shows no differences between groups A, B, and C, but how do these compare regionally (e.g., buccal vs. lingual gaps) – were subgroup analyses performed?

Response: Thank you. We have added the 95% confidence intervals for each reported parameters in result part. We also presented the gaps in each measured points (lingual, buccal, mesial and distal) as your suggestion into Table 3 and Table 4 with their interpretation.

9. Table 1 uses superscript letters for similarity, but what p-values correspond to these comparisons, and was homogeneity of variance confirmed for ANOVA?

Response: Thank you. We added the p-value and test results into Table 2. All the variables were checked for their distribution and homogeneity before using Anova.

10. Discussion

Here I suggest that you more critically approach limitations and alternative explanations. Given the inferior fit in Dentbird's automatic mode, what software-specific factors (e.g., algorithm sensitivity to curvature) might explain this, and how could future updates address it?

Response: Thank you for your suggestion. However, the software algorithms has not been published.

11. The study cites clinical acceptability (<120 µm), but how do these gaps compare to in vivo studies where oral fluids or cementation might alter fits?

Again, is this level of adaptation really acceptable today?

Response: Thank you. Might we answer this question as above?

12. Limitations mention in vitro constraints, but why not discuss potential biases from using a metal typodont (e.g., vs. natural tooth reflectance in scanning)? Your study is confined to a single tooth type (mandibular first molar) with a chamfer finish line, limiting generalizability. No evaluation of horizontal gaps or more complex clinical scenarios. Please comment.

Response: Thank you so much for these questions. Using metal typodont, only doing on lower first molar typodont, no horizontal evaluation and only chamfer lines are the limitations of our study. We did discussed about these into Lines 227 – 241, page 13.

13. Conclusion

How might these results influence the choice between free and paid software in educational or low-resource settings?

Response: As the results of this study, Dentbird automatic finishing lien detection is useful and acceptable to design an interim crown. It might be one of the cheapest, fastest and easiest way for temporary crowns with acceptable fit.

14. The emphasis on semi-automatic equivalence between software is strong, but what training implications does this have for operators relying on automation?

Response: For the automatic process, the most important step is selecting one or two guidance points on the finishing line. Guidance points are the points where the curvature change is greatest.

15. Figures

Fig. 1 could show different views of the prepared tooth. Fig. 2, as presented, lacks the resolution that allows for better viewing of the software screens.

Response: We have changed figure 1 with different views of the typodont. Figure 2 has been changed to the higher resolution version.

16. References

All references must adhere to PLOS ONE standards. Please check carefully.

Some references may be incomplete.

Response: Thank you.

17. Language

There are some typos and inconsistencies through the text, which deserve careful review.

Response: Thank you.

---

## [Decision Letter · Decision Letter 1]

28 Dec 2025

Evaluation of the Marginal and Internal Gaps in 3D-Printed Interim Crowns Using Different Finish Line Detection Methods: An In Vitro Study

PONE-D-25-09341R1

Dear Dr. Nguyen,

We’re pleased to inform you that your manuscript has been judged scientifically suitable for publication and will be formally accepted for publication once it meets all outstanding technical requirements.

Kind regards,

Nour Ammar

Academic Editor

PLOS One

Additional Editor Comments (optional):

Reviewers' comments:

Reviewer's Responses to Questions

**Comments to the Author**

Reviewer #2: All comments have been addressed

Reviewer #3: All comments have been addressed

2. Is the manuscript technically sound, and do the data support the conclusions?

Reviewer #2: Yes

Reviewer #3: Yes

3. Has the statistical analysis been performed appropriately and rigorously?

Reviewer #2: Yes

Reviewer #3: Yes

4. Have the authors made all data underlying the findings in their manuscript fully available?

Reviewer #2: Yes

Reviewer #3: No

5. Is the manuscript presented in an intelligible fashion and written in standard English?

Reviewer #2: Yes

Reviewer #3: Yes

Reviewer #2: I am satisfied my comments/suggestions have been adressed.

Reviewer #2:

1.Introduction: There are previous studies (ADD MORE RECENT REFERENCES) that

evaluated the quality of restorations designed by digital software such as CEREC (13-

15), Lava (13, 16), EXOCAD (12). The results of these studies show that the

restorations designed by CAD/CAM software have acceptable marginal adaptability.

2.References: Incorporate 2 or more recent references as mentioned above.

Response: Thank you. We have added more recent papers evaluating the quality of

crowns designed by charged CAD software (Line 66, page 11)

Reviewer #3: Thanks for the responses to my comments and questions.

Also for the modifications in the manuscript text.

**Do you want your identity to be public for this peer review?** For information about this choice, including consent withdrawal, please see our Privacy Policy

Reviewer #2: No

Reviewer #3: No

---

## [Editor Report · Acceptance letter]

PONE-D-25-09341R1

PLOS One

Dear Dr. Nguyen,

I'm pleased to inform you that your manuscript has been deemed suitable for publication in PLOS One. Congratulations! Your manuscript is now being handed over to our production team.

Kind regards,

on behalf of

Dr. Nour Ammar

Academic Editor

PLOS One